# Seed Germination after 30 Years Storage in Permafrost

**DOI:** 10.3390/plants9050579

**Published:** 2020-05-02

**Authors:** Svein Øivind Solberg, Guro Brodal, Roland von Bothmer, Eivind Meen, Flemming Yndgaard, Christian Andreasen, Åsmund Asdal

**Affiliations:** 1Faculty of Applied Ecology, Agricultural Sciences and Biotechnology, Inland Norway University of Applied Sciences, NO-2418 Elverum, Norway; 2Norwegian Institute of Bioeconomy Research, NIBIO, P.O. Box 115, NO-1431 Ås, Norway; guro.brodal@nibio.no; 3The Faculty of Landscape Planning, Horticulture and Agricultural Sciences has Six Departments, Swedish University of Agricultural Sciences, SE-230 53 Alnarp, Sweden; roland.von.bothmer@slu.se; 4Kimen Seed Laboratory, P.O. Box 164, NO-1431 Ås, Norway; eivind.meen@kimen.no; 5Nordic Genetic Resource Center, Box 41, SE-230 53 Alnarp, Sweden; flemming@yndgaard.se (F.Y.); asmund.asdal@nordgen.org (A.A.); 6Department of Plant and Environmental Sciences, Copenhagen University, Højbakkegaard Allé 13, DK2630 Taastrup, Denmark; can@plen.ku.dk

**Keywords:** ex situ conservation, germination, longevity, plant genetic resources, seed storage

## Abstract

More than 30 years ago, the Nordic Gene Bank established a long-term experiment on seeds stored under permafrost conditions in an abandoned mine corridor in Svalbard, as a tool to monitor storage life under these conditions. The study included seeds from 16 Nordic agricultural and horticultural crops, each represented by two or three cultivars (altogether 38 accessions). All seeds were ultra-dried to 3–5% moisture before being sealed in glass tubes. Germination tests were performed in accordance with the International Seed Testing Association (ISTA) protocols. At the initiation of the experiment, the samples showed good germination with the median value at 92%. The overall picture remained stable over the first twenty to twenty-five years. However, the variation became larger over time and at 30 years, the median value had dropped to 80%. At the lower end, with a high drop in germination, we found rye, wheat, and English ryegrass. At the upper end, we found Kentucky bluegrass and cucumber. The lowest germination was found in samples with the highest initial seed moisture levels. Pre-storage conditions are likely to be of major importance for longevity.

## 1. Introduction

Most food plants produce seeds that can be stored under low temperature and moisture conditions. Much of our knowledge on seed longevity is based on artificial ageing experiments, where seeds are exposed to suboptimal conditions of elevated temperature and moisture for some weeks, and storage life is predicted based on the seed moisture content, storage temperature, and seed lot characters [1,2]. Such calculations have predicted that high-quality seeds could survive ideal conditions for hundreds of years or more [3,4], which was good news for many gene banks, but might be unrealistic as such studies have rarely been confirmed in long-term storage studies. Long-term seed storage is crucial for ex situ gene bank conservation [5,6,7]. Gene banks maintain crop diversity and facilitate the utilization of seeds for breeding, research, education, and other purposes [8,9,10,11]. Safety back-ups are kept, ideally at a second location, to spread the risks [12,13]. The Global Seed Vault at the Arctic Archipelago of Svalbard was opened in 2008 and is facilitating such back-up collections with a world outreach [14]. However, more than twenty years before this, the Nordic Gene Bank (NGB) started a small seed storage facility in an abandoned coalmine corridor in Svalbard. Different options had been considered, such as inland ice caves in Greenland or mountain caves in Jotunheimen, Norway, but in the end, a coalmine in the permafrost in Svalbard was chosen due to its good logistics, despite its remote location [15]. The temperature of the Global Seed Vault in Svalbard *was* −18 °C compared to the −3.5 °C present in the abandoned coalmine corridor. The rock temperature was stable, which meant that it was independent of an external energy supply. As a tool to monitor storage life under these permafrost conditions, a long-term seed storage experiment was initiated. The experiment started in 1986 and included samples for germination monitoring until 2086, thus it was termed “the 100 year experiment”. Important crops for Nordic agriculture and horticulture were included. The investigation is still ongoing, and in this paper, we summarize the results after the first 30 years.

## 2. Results and Discussion

### 2.1. Overall Patterns

The overall picture of seed germination development over storage time across all accessions (species and cultivars) is illustrated by boxplots (Figure 1). A lower and upper percentile defines the box, in which 75% of the observations were found. A marked line denotes the median germination value, and the whiskers and small circles show observations away from the box. At the initiation of the experiment in 1986 (year 0 = y0), most of the seed lots showed an excellent germination ability. The box-range was from 83% to 95%, and the median was at 92% germination. The overall picture was relatively stable over the first twenty to twenty-five years, but the variation increased over time as some seed lots showed a reduced germination. After 30 years, the median value across all the lots was 80%, but with outlier samples below 40%, and some of the lots in the 50–70% range. 

Figure 2 shows a dendrogram of a cluster analysis of the germination results of the accessions. The dissimilarity values of the fusion level values of the dendrogram indicated the cutting level six clusters to be correct. The largest cluster, cluster 1, contained 19 lots. It contained both normal and lots with seed-borne pathogens (i-prefix lots). Also, cluster 2 contained both normal and seed-borne pathogens, as did cluster 6. In cluster 1 we found Barley 1, Ryegrass 1, Ryegrass 2, Timothy 2, Bluegrass 2, Redclover 2, Rape 1, Rape 2, Onion 1, Onion 2, Carrot 1, Carrot 2, Cauliflower 2, i1 Wheat, i2 Wheat, i Barley, i Meadow_fescue, i Onion, and i Cabbage. In cluster two, we found Barley 2, Bluegrass 1, Beet 2, Lettuce, Cabbage, Cucumber 1, Cucumber 2, Cauliflower 1, and i-Timothy. In cluster three, we found Wheat 1, Timothy 1, Redclover 1, and i Carrot. In cluster 4, we found Wheat 2. In cluster 5, we found Rye 1 and Rye 2. In cluster 6, we found Beet 1, i Lettuce, and i Beet. Lots with seed-borne pathogens (i-prefix lots) were spread over the clusters, showing that such pathogens do not explain much of the seed longevity. Furthermore, seed lots of the same species were only partly in the same clusters; for example, the two cucumber (*Cucumis sativus* L.) lots and the two rye (*Secale cereale* L.) lots. This also shows that interspecies differences are not of importance for explaining seed longevity. 

The seeds of all the species remained viable after 30 years in permafrost under the given conditions (dried to 3–5% moisture content and stored in sealed glass ampoules). Crops, but also cultivars, within the same crops showed different results. Similar patterns have also been observed in other long-term experiments, both under ambient [16,17] or −18 °C conditions [18,19,20,21]. The cold storage of seeds should provide improved longevity compared to ambient storage [22]. Another factor of importance is seed maturation [23,24].

For instance, the potential seed longevity of barley was found to be best if it was harvested a week or so after grain filling was completed [25]. The same has been found in tomatos (*Solanum lycopersicum* L. var. *lycopersicum*) [26]. Furthermore, weather, seed coat damages, diseases, and pests may influence seed storage life [24,27]. In our experiment, the samples were all from newly harvested seeds, but we do not have data on the weather or other pre-harvest conditions. We only assume that the seed lots selected for the experiment were of a high quality.

### 2.2. Crop-Wise Results

Our results show good performance for many of the vegetables, while the picture is more varied for cereals and forage species (Appendix A). Table 1 gives the germination result for each seed lot. According to the FAO’s gene bank standard [13], “regeneration shall be carried out when the viability drops below 85 percent of the initial viability or when the remaining seed quantity is less than what is required for three sowings of a representative population of the accession.” Seed lots with the highest loss in germination, here defined as a loss in 15% or more over the 30 years, were found in both of the rye lots, two of the three English ryegrass (*Lolium perenne* L.) lots (Ryegrass 1, Ryegrass 2), two out of four wheat (*Triticum aestivum* L.) lots (Wheat 1, Wheat 2), one of the three timothy (*Phleum pratense* L.) lots (Timothy 1), one of the three barley (*Hordeum vulgare* L.) lots (Barley 1), and in the one seed lot of meadow fescue (*Schedonorus pratensis* (Huds.) P. Beauv.). The literature shows that especially rye, but also some forage grasses, can be relatively short-lived [28,29,30,31]. Intermediate storage performance, with a 5–15% loss in germination over the 30-year period, was found in two of the three lettuce (*Lactuca sativa* L.) lots (i Lettuce, Lettuce 2), two of the three carrot (*Daucus carota* subsp. *sativus* (Hoffm.) Schübl & G. Martens) lots (Carrot 2, i-Carrot), one of the four wheat lots (i1-Wheat), one of the three barley lots (Barley 2), one of the three timothy lots (Timothy 2), one of the two cauliflower (*Brassica oleracea* L. var. *botrytis*) lots (Cauliflower 2), one of the two oilseed rape (*Brassica napus* L.) lots (Rape 1), and one out of the three red clover (*Trifolium pratense* L.) lots (Redclover 1). The most long-lived, with a loss in germination less than 5% after 30 years of storage, were found in all of the three beet (*Beta vulgaris* L.) lots, all of the three onion (*Allium cepa* L.) lots, both of the cucumber lots, both of the Kentucky bluegrass (*Poa pratensis* L.) seed lots, the only cabbage (*Brassica oleracea* L.) seed lot, and in the last of the red clover, cauliflower, barley, wheat, oilseed rape, carrot, and lettuce seed lots. Other studies have also shown that beet seeds as well as cucumber seeds have retained a high germination level over time, but, in contrast to our study, onions are generally found to be short-lived [17,18,19,29,30,31]. Our results are a little surprising as we found that onion, and to some extent lettuce, showed no decline in germination over the 30-year period.

### 2.3. Moisture Measurements

For all samples, the variation between the highest and lowest moisture content over the ten years was between 0.3% and 0.8% (Table 2). The data showed that two lots had a higher moisture content exceeding 5% in the initial test; these were the samples Wheat 2 ‘Solid’, with 6.3% humidity, and Rye 1 ‘Petkus II’, with 5.3% humidity. For wheat, one seed lot showed a low decline, and two seed lots showed a steep decline in germination. Different pre-harvest conditions or genetic factors may explain a steep decline in germination. We saw an effect of drying the seeds to lower than 5% internal moisture content before packing. The two samples with the highest internal humidity (Wheat 2 and Rye 1) were among the ones that showed the most significant drop in germination in our experiment. The current FAO standards [13], which are used by most gene banks, recommend drying for three months at 15 °C and 15% RH. According to our experience, this would give a seed moisture content exceeding 5%, thus decreasing the longevity. 

## 3. Conclusions

The study has so far revealed valuable results concerning the longevity of seeds after 30 years in permafrost. Nine out of the 38 seed lots showed a germination loss exceeding 15%, which is the level recommended by the FAO for carrying out regeneration. Rye and ryegrass in particular showed a rapid decline, while many of the vegetables showed a low decline in germination. The results are relevant for the seeds in the first Nordic back-up collection stored in the abandoned coalmine. A given storage condition is an essential characteristic for comparing results with other experiments. Here, seeds were stored in permafrost with a stable sub-zero temperature. We had no reference material at −18 °C conditions. Despite the limited number of samples and the lack of a −18 °C control, the observations add knowledge about the longevity of seeds.

## 4. Materials and Methods 

### 4.1. Seed Samples and Seed Storage

In total, 38 seed lots covering 16 crops were included in the study (Table 3). Each species was represented by two to four seed lots, except for cabbage and meadow fescue, which had only one. The seed lot ID was given by crop name and code. A number after the crop name (1 or 2) refers to seed lots where we have no information on seed-borne diseases. A prefix “i” letter applies to seed lots where we know that seed-borne diseases were present. 

The lots were from different cultivars and origins. No cultivation details are available except that the seed lots should be of good quality. Initial germination tests were conducted (y0). Before storage, all the seeds were dried to 3–5% internal moisture content using a Munters (Kista, Sweden) dehumidifier adjusted to 10% RH in a room at 25 °C. After that, they were placed in a freezer overnight before they were sealed in glass ampoules. Each seed lot was divided into sub-samples of 2 × 500 seeds for each withdrawal. The sub-samples were boxed and labelled with the date for withdrawal, transported to Longyearbyen, Svalbard, and placed in the abandoned coalmine corridor. The temperature inside the coalmine was measured as −3.5 °C (± 0.2 °C).

### 4.2. Germination Studies

The first set of samples was tested in December 1986 (year 0), with a plan to have the last tests in 2086 (year 100). Since the start of the experiment, 11 boxes have been retrieved. The moisture content of the seed was monitored from year 0 to year 10 (Table 4). The germination tests and the moisture content tests were carried out at the Kimen Seed Laboratory (Ås, Norway). The testing followed the International Seed Testing Association (ISTA) rules [32,33]. The details of the germination conditions for the different species in the study are shown in Table 5. In the case of oilseed rape and onion, the number of days to final count was, however, larger than the number of days in the ISTA protocols. In accordance with the protocols, we germinated 4x100 or 8x50 seeds per sample and each replicate was compared to the mean. If a large variation was detected, the samples were re-tested. Different types of filter paper (Seedburo Equipment Co, Des Plaines, IL, USA) were used as substrates for the germination tests. 

For cereals, in-between paper methods (BP) were used; the seeds were placed on one moist paper with a second paper on top, and thereafter rolled and placed vertically in plastic. Inside the roll, the seeds germinated, and the seedlings developed. For grasses, red clover, and vegetables (except for beet), a Jacobsen apparatus was used [34], which is a plate where circular pieces of wet filter paper and seeds are placed and kept moist by a wick (TP). The seeds of beet were germinated between humidified pleated filter paper. The germination temperature varied from alternating between +20 °C for 16 h and +30 °C for 8 h to constant ambient temperatures. Lettuce required light for germination, but for the other species light was not applied. Tests of actual moisture content were made along with seed germination tests in year 0 and the first 10 years (Table 2). 

### 4.3. Data Analyses

R software was used for statistical examination. One value was missing for year 5 for red clover (cultivar Jokioinen). It was replaced with the average of the four nearest neighbor values: 76.0%. The short descriptor names of the eleven test results are year 0, year 2.5, year 5, year 7.5, year 10, year 12.5, year 15, year 17.5, year 20, year 25, and year 30, where year 0 represents the results of the trial that started in 1986. Summary statistics and boxplots were used to overview the data. The R function ‘time series’ was used to illustrate the fluctuation of the results over the 11 test occasions over the first 30 years. We calculated the loss in germination (Δ germ) by averaging the germination percentage of the three first test occasions (y0, y2.5, and y5) minus the test occasion at y30. We then categorized the samples into three: the most short-lived group (with Δ germ of 15% or more, which is at a level where regeneration should have been carried out), the intermediate group (Δ germ 5% to 15%), and the most long-lived (Δ germ less than 5%, or with less than 95% germination loss over the 30 years).

## Figures and Tables

**Figure 1 plants-09-00579-f001:**
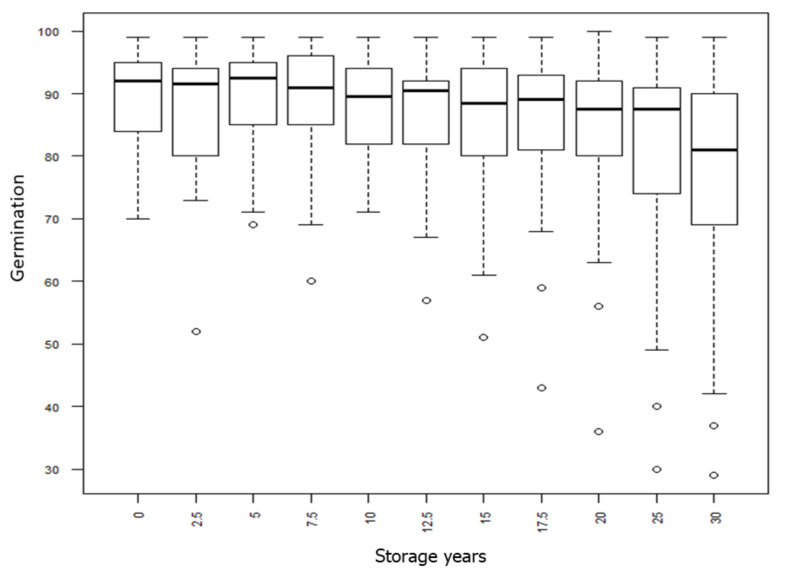
Boxplots showing the germination percentages across species and cultivars throughout the first thirty years (year 0, year 2.5, year 5, year 7.5, year 10, year 12.5, year 15, year 17.5, year 20, year 25, and year 30).

**Figure 2 plants-09-00579-f002:**
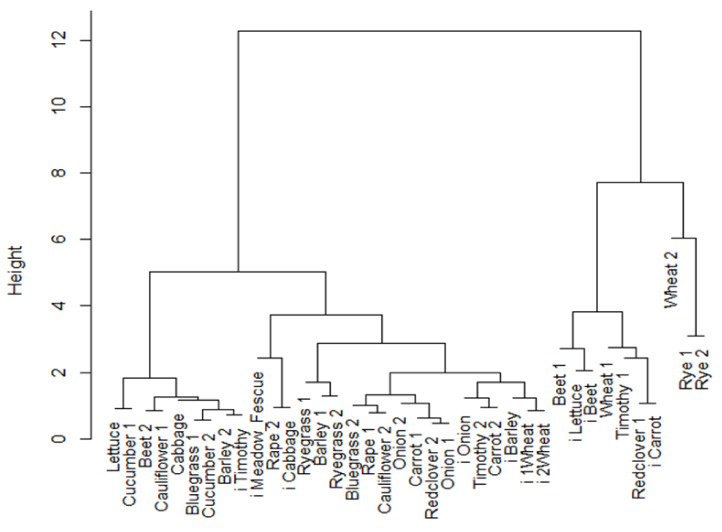
The dendrogram of a cluster analysis of the germination results of the seed lots.

**Table 1 plants-09-00579-t001:** Germination percentages for all the seed lots (values below 70% in bold), and loss in germination (Δ germ) over the first 30 years, calculated by averaging the germination percentage of the three first test occasions (year 0, year 2.5, and year 5) minus the last test occasion (year 30).

			Storage Years									
Seed Lot ID	0	2.5	5	7.5	10	12.5	15	17.5	20	25	30	Δ Germ
Barley 1	95	96	96	96	95	91	95	95	79	90	76	−20%
Barley 2	95	94	96	97	94	91	97	94	93	92	86	−9%
i Barley	90	89	94	94	92	82	89	88	85	85	88	−3%
Wheat 1	75	75	83	83	80	85	77	76	71	**58**	**57**	−21%
Wheat 2	70	**52**	89	89	82	86	79	76	**64**	**40**	**37**	−33%
i1 Wheat	89	85	89	89	88	84	87	87	87	82	80	−8%
i2 Wheat	90	83	91	93	89	87	85	89	84	83	87	−1%
Rye 1	76	74	84	84	74	**57**	**51**	**43**	**36**	**30**	**29**	−49%
Rye 2	81	78	87	83	74	**67**	**61**	**59**	**56**	**49**	**48**	−34%
Ryegrass 1	99	96	96	96	95	93	93	88	85	**74**	**59**	−38%
Ryegrass 2	96	95	97	95	96	93	94	92	94	87	**69**	−27%
Timothy 1	73	77	77	**69**	79	71	73	**68**	**63**	**63**	**42**	−34%
Timothy 2	90	91	93	89	90	92	88	92	85	81	78	−13%
i Timothy	94	95	92	96	92	94	95	92	94	91	92	−2%
Bluegrass 1	92	95	95	97	97	96	95	94	92	93	92	−2%
Bluegrass 2	94	91	89	96	93	91	94	90	92	89	88	−3%
iMeadow fescue	92	90	89	86	88	90	87	81	81	**69**	**54**	−36%
Red clover 1	75	74	76	80	75	74	**68**	76	**68**	73	**68**	−7%
Red clover 2	93	94	94	92	89	91	89	89	90	91	90	−4%
Beet 1	79	79	71	**60**	76	72	75	70	80	70	78	0%
Beet 2	97	97	97	97	97	98	94	95	95	98	97	0%
i Beet	78	78	69	77	81	81	80	82	82	79	77	0%
Rape 1	95	94	95	96	90	91	90	90	92	91	87	−8%
Rape 2	84	84	83	87	85	84	84	82	88	82	81	−3%
Onion 1	92	92	92	93	90	89	89	88	91	90	91	−1%
Onion 2	89	94	92	92	91	94	92	94	90	92	89	−3%
i Onion	92	93	87	87	88	90	85	91	87	88	88	−3%
Lettuce 1	98	95	98	98	98	92	99	98	98	99	99	0%
Lettuce 2	97	93	96	90	94	97	93	93	92	92	90	−5%
i Lettuce	84	80	78	80	76	71	78	77	73	75	74	−7%
Cucumber 1	99	99	99	99	99	99	99	99	100	98	98	−1%
Cucumber 2	93	94	94	95	97	92	96	95	92	92	90	−4%
Carrot 1	92	90	93	90	89	91	86	89	88	90	91	−1%
Carrot 2	91	93	94	86	88	90	85	89	90	89	81	−12%
i Carrot	76	73	72	74	71	75	**67**	78	**65**	71	**68**	−6%
Cauliflower 1	95	93	94	98	98	98	98	94	96	94	91	−3%
Cauliflower 2	95	92	94	95	94	92	90	92	93	90	79	−15%
i Cabbage	87	82	85	85	86	82	86	82	80	76	80	−5%

**Table 2 plants-09-00579-t002:** Seed moisture content (in %) in the different lots over the first ten years of the experiment.

		Storage Years				
Seed Lot	0	2.5	5.0	7.5	10	Max/Min
Barley 1	5.0	4.7	4.6	4.7	4.8	5.0/4.6
Barley 2	4.6	4.6	4.3	4.3	4.7	4.7/4.3
Wheat 1	4.0	4.3	4.3	4.3	4.3	4.3/4.0
Wheat 2	6.3	5.8	5.5	5.8	5.8	6.3/5.5
Rye 1	5.3	5.1	4.9	5.1	5.2	5.3/4.9
Rye 2	4.9	4.7	4.4	4.7	4.8	4.9/4.4
Beet 1	3.7	4.3	4.3	3.7	4.1	4.3/3.7
Beet 2	3.5	3.7	3.6	3.9	3.5	3.9/3.5
Cucumber 1	3.0	3.5	3.6	2.8	2.9	3.6/2.8
Cucumber 1	2.8	3.3	2.7	2.8	2.7	3.3/2.7

**Table 3 plants-09-00579-t003:** Overview of the examined crops and sample information. A prefix number refers to the samples with no information on seed-borne diseases, while a prefix i-letter refers to the samples where we detected seed-borne diseases to be present at the start of the experiment.

Crop/Species	Sample ID and Cultivars (Country of Origin)
Barley (*Hordeum vulgare*)	1 = Inga Abed (DNK), 2 = Tunga (NOR), i = Bamse (NOR)
Wheat (*Triticum aestivum*)	1 = Vakka (FIN), 2 = Solid (SWE), i1 = Runar (NOR), i2 = Line 79 CBW A72 (CAN)
Rye (*Secale cereale*)	1 = Petkus II (DNK), 2 = Voima (DNK)
English ryegrass (*Lolium perenne*)	1 = Pippin (DNK), 2 = Riikka (FIN)
Timothy (*Phleum pratense*)	1 = Tammisto (FIN), 2 = Bodin (NOR), I = Forus (NOR)
Kentucky bluegrass (*Poa pratensis*)	1 = Hankkijan Kyösti (FIN), 2 = Annika (DNK)
Meadow fescue (*Schedonorus pratensis*)	I = Salten (NOR)
Red clover (*Trifolium pratense*)	1 = Jokioinen (FIN), 2 = Molstad (NOR)
Beet (*Beta vulgaris*)	1 = 311 N typ (SWE), 2 = 70500 (DNK)^2^, i = Hilma (GBR)
Oilseed rape (*Brassica napus*)	1 = Jupiter (SWE), 2 = Linrama (DNK)
Onion (*Allium cepa*)	1 = Hamund (SWE), 2 = Owa (DNK), i = Laskala (NOR)
Lettuce (*Lactuca sativa*)	1 = Attraktion Sana (DNK), 2 = Hilro (SWE), i = Attractie (NLD)
Cucumber (*Cucumis sativus*)	1 = Langelands gigant (DNK), 2 = Rhensk Druv (SWE)
Carrot (*Daucus carota*)	1 = Nantes Fancy (DNK), 2 = Regulus (SWE), I = Forto Nantes (NLD)
Cauliflower (*B. oleracea* v. *botrytis*)	1 = Svavit (SWE), 2 = Pari (DNK)
Cabbage (*Brassica oleracea*)	I = Trønder Lunde (NOR)

**Table 4 plants-09-00579-t004:** Dates of the retrieval of the samples over the first 30 years, and types of tests conducted.

Year	Date	Germination	Moisture
0	1986, December	X	X
2.5	1989, June	X	X
5	1991, December	X	X
7.5	1994, June	X	X
10	1996, December	X	X
12.5	1999, June	X	
15	2001, December	X	
17.5	2004, June	X	
20	2006, December	X	
25	2011, December	X	
30	2016, December	X	

**Table 5 plants-09-00579-t005:** Germination conditions for the different crops included in the experiment.

Crop/Species	Germination Substrate	Temperature (°C)	Final Count (Day Number)
Barley	BP	20	7
Wheat	BP	20	8
Rye	BP	20	7
English ryegrass	TP	20<=>30	14
Timothy	TP	15<=>25	10
Kentucky bluegrass	TP	15<=>25	28
Meadow fescue	TP	20<=>30	14
Red clover	TP	20	10
Beet	PP	20	14
Oilseed rape	TP	20	10^3^
Onion	TP	15	14^4^
Lettuce	TP	20-light	7
Cucumber	TP	20<=>30	8
Carrot	TP	20	14
Cauliflower, cabbage	TP	20<=>30	10

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
