# Peer review of "Seed Germination after 30 Years Storage in Permafrost"

_plants, 2020, doi:10.3390/plants9050579_

Round 1

Reviewer 1 Report

In the manuscript entitled “Seed viability after 30 years storage in permafrost” the authors tested seed germination from samples that have been stored under permafrost conditions in an abandoned mine corridor at Svalbard for 30 years. The study is set to test seeds stored for 100 years.

The importance of this work is high because we need to test seed viability capacity for germination upon a long period of time under cold conditions. The establishment of the Global Seed Vault at the Arctic Archipelago Svalbard will only serve its purpose if the stored material could germinate. This kind of work that requires a long-life span should be praised and cherished.

Both material and methods and results are clear and well organized and are easy to follow.

I have a small comment that could be mentioned in the introduction;

What is the temperature of the Global Seed Vault in Svalbard compared with the -3.5°C present in the abandoned coalmine corridor at Svalbard, place where the seeds for this study are stored?

Author Response

Feedback to reviewer 1

“I have a small comment that could be mentioned in the introduction; What is the temperature of the Global Seed Vault in Svalbard compared with the -3.5°C present in the abandoned coalmine corridor at Svalbard, place where the seeds for this study are stored?

Our response:

We added a sentence in the introduction describing this

The temperature of the Global Seed Vault in Svalbard is −18°C compared with the -3.5°C present in the abandoned coalmine corridor.

Reviewer 2 Report

The presented manuscript is based on analysis of a unique resource of 30 year old seed stored in permafrost conditions. The manuscript is clearly written and presents the observations made in the context of literature. I believe this is an interesting manuscript that could contribute to ongoing discussion on seed storage. It is a bit disappointing as to the low level of data acquired. I do understand seed viability was the primary target but I would think some more advance analysis including physiological measurements like days to radical protrusion, days to seedlings establishment, etc. I understand it is now to late as the past germination tests are already done but it maybe a good idea to set a side some of the material in the future teste for other analysis including molecular assessments. – who know what will be available in another 30 years.

 As for the presented analysis. Figure 2 heatmap shows surprising low clustering. I agree with authors interpretation that there is probably no clear correlation within species. But maybe the authors could extend their analysis and include a dendrogram of delta in lot viability ? Also given that the authors do have some data on moisture contents it would be probably possible to conduct some type of more advanced statistical multivariant  analysis based on anova correlation – but I am not a statistician.

The quality of on figure description in Figure 2 is too low to read.

 line 154 " a–18°oC control,"  double °o

Author Response

Feedback to reviewer 2

1

“ It is a bit disappointing as to the low level of data acquired. I do understand seed viability was the primary target but I would think some more advance analysis including physiological measurements like days to radical protrusion, days to seedlings establishment, etc. I understand it is now to late as the past germination tests are already done but it maybe a good idea to set a side some of the material in the future teste for other analysis including molecular assessments. – who know what will be available in another 30 years.”

Our response:

Thank you for this good comment. For 2017 we included data for first count, which is an indicator for seed vigour. However, as it was only made in this last test, we did not include it in the paper. Certainly, it will be a point to follow up in future tests.

2

"As for the presented analysis. Figure 2 heatmap shows surprising low clustering. I agree with authors interpretation that there is probably no clear correlation within species. But maybe the authors could extend their analysis and include a dendrogram of delta in lot viability ? Also given that the authors do have some data on moisture contents it would be probably possible to conduct some type of more advanced statistical multivariant analysis based on anova correlation – but I am not a statistician".

"The quality of on figure description in Figure 2 is too low to read."

Our response:

We changed Figure 2 to a simple cluster illustration. We changed the heading of the figure accordingly and removed the sentence about heatmat in M&M. In the results we say:

Figure 2 shows a dendrogram of a cluster analysis of the germination results of the accessions. The dissimilarity values of the fusion level values of the dendrogram indicated the cutting level six clusters to be correct. The largest cluster 1 contained 19 lots. It contained both normal and lots with seed-borne pathogens (i-prefix lots). Also cluster 2 contained both normal and seed-borne pathogens. The same did cluster 6….

3

“line 154 " a–18°oC control,"  double °o”

Thanks, we have corrected this error.